# Targeting E3 Ubiquitin Ligases and Deubiquitinases in Ciliopathy and Cancer

**DOI:** 10.3390/ijms21175962

**Published:** 2020-08-19

**Authors:** Takashi Shiromizu, Mizuki Yuge, Kousuke Kasahara, Daishi Yamakawa, Takaaki Matsui, Yasumasa Bessho, Masaki Inagaki, Yuhei Nishimura

**Affiliations:** 1Department of Integrative Pharmacology, Graduate School of Medicine, Mie University, Tsu, Mie 514-8507, Japan; tshiromizu@doc.medic.mie-u.ac.jp (T.S.); 320d030@m.mie-u.ac.jp (M.Y.); 2Department of Physiology, Graduate School of Medicine, Mie University, Tsu, Mie 514-5807, Japan; kkasahara@doc.medic.mie-u.ac.jp (K.K.); dyama@doc.medic.mie-u.ac.jp (D.Y.); minagaki@doc.medic.mie-u.ac.jp (M.I.); 3Gene Regulation Research, Division of Biological Sciences, Nara Institute of Science and Technology, Takayama, Nara 630-0192, Japan; matsui@bs.naist.jp (T.M.); ybessho@bs.naist.jp (Y.B.)

**Keywords:** ubiquitin–proteasome pathway, cilia, ciliogenesis, differentiation, proliferation, ciliopathy, cancer

## Abstract

Cilia are antenna-like structures present in many vertebrate cells. These organelles detect extracellular cues, transduce signals into the cell, and play an essential role in ensuring correct cell proliferation, migration, and differentiation in a spatiotemporal manner. Not surprisingly, dysregulation of cilia can cause various diseases, including cancer and ciliopathies, which are complex disorders caused by mutations in genes regulating ciliary function. The structure and function of cilia are dynamically regulated through various mechanisms, among which E3 ubiquitin ligases and deubiquitinases play crucial roles. These enzymes regulate the degradation and stabilization of ciliary proteins through the ubiquitin–proteasome system. In this review, we briefly highlight the role of cilia in ciliopathy and cancer; describe the roles of E3 ubiquitin ligases and deubiquitinases in ciliogenesis, ciliopathy, and cancer; and highlight some of the E3 ubiquitin ligases and deubiquitinases that are potential therapeutic targets for these disorders.

## 1. Introduction

Cilia are antenna-like structures that are present in a variety of vertebrate cells [1,2,3,4,5,6]. There are two broad classes of cilia: Motile and nonmotile cilia [1]. The nonmotile cilia are called primary cilia. Both motile and primary cilia contain receptors and channels that detect signals from extracellular cues, such as mechanical flow and chemical stimulation, and transduce them into the cell, where they contribute to the maintenance of proper development and homeostasis. Considering these functions, it is not surprising that dysregulation of cilia function can cause cancer and other diseases, including ciliopathies, which manifest as various disease phenotypes, such as congenital anomalies, neurodevelopmental disorders, and obesity [1,6,7,8,9].

The structure and function of cilia are dynamically and precisely regulated, enabling cells to proliferate, migrate, and differentiate in a spatiotemporally controlled manner [6,10]. The primary cilium is composed of three compartments: The basal body, the transition zone, and the axoneme [2]. The basal body is derived from the mother centriole. Both centrioles and basal body contain nine circularly arranged triplets of microtubules (A-, B-, and C-tubules). The axoneme consists of nine microtubule doublets projected from the A- and B-tubules of the basal body. A central pair of singlet microtubules is present (+2) and absent (+0) in motile and primary cilia, respectively [11,12,13,14]. Therefore, the axoneme of motile and primary cilia is described as 9 × 2 + 2 and 9 × 2 + 0, respectively. The transition zone is a short area located above the basal body characterized by Y-shaped fibers connecting the microtubule doublets to the ciliary membrane [15]. The structure of motile cilia is more complex [16]. The daughter centriole plays an important role in the formation of motile cilia [16,17].

Primary cilia are disassembled and assembled when cells enter mitosis and exit the cell cycle, respectively [18,19,20]. The formation of primary cilia starts with the binding of small cytoplasmic vesicles transported from the Golgi apparatus to the mother centriole and conversion from the mother centriole to the basal body. The basal body is then moved and anchored to the plasma membrane. Coiled-coil protein 110, a component of the inhibitory complex of ciliogenesis, is removed to initiate axoneme elongation [21,22]. The ciliary vesicle then fuses with the plasma membrane, and large amounts of tubulin are transported from the cytoplasm into the cilium to extend the axoneme [23]. Many signaling molecules are also transported from the cytoplasm into the cilium (anterograde) and from the cilium into the cytoplasm (retrograde) by kinesin and dynein, respectively, which are motor proteins that travel along the axoneme [24].

Various types of posttranslational modification, including phosphorylation, acetylation, and ubiquitination, are involved in the dynamic regulation of the structure and function of cilia [2,4,5,6,25]. Modification of proteins by attachment of ubiquitin, a highly conserved 76-amino-acid protein, is a critical step in targeting the selective degradation of proteins by proteasomes as part of the ubiquitin–proteasome system (UPS) [26]. Protein ubiquitination occurs in three steps. First, ubiquitin-activating enzymes (E1) bind to ubiquitin, which is expressed in all cell types; second, ubiquitin is transferred from E1 enzymes to ubiquitin-conjugating enzymes (E2); and finally, ubiquitin-ligating enzymes (E3) transfer the ubiquitin from E2 enzymes and ligate it to lysine residues on the target protein. To date, 2, approximately 40, and about 600 E1, E2, and E3 enzymes, respectively, have been identified in humans [26]. The selectivity of target protein ubiquitination is conferred by the combination of E2 and E3 enzymes. Protein ubiquitination is counteracted by deubiquitinase (DUB)-mediated removal of ubiquitin moieties from ubiquitinated proteins [27]. About 100 DUBs have been identified in humans. The balance between ubiquitination and deubiquitination of target proteins and their proteasomal degradation are tightly regulated processes, and dysregulation of the UPS has been detected in various disorders [28,29,30].

Several lines of evidence support a major role for the UPS in regulating the structure and function of cilia [4,5,6,31,32,33,34,35]. Here, we briefly review the role of cilia in phenotypes of ciliopathy and cancer. We then focus on the role of E3 ubiquitin ligases and DUBs in ciliogenesis, ciliopathy, and cancer, and suggest that these enzymes may serve as novel therapeutic targets for the development of treatments for these disorders.

## 2. Roles of Cilia in Ciliopathy and Cancer

Cilia play crucial roles in the development of vertebrates. In some cell types, cilia are present only transiently during a critical point in development [36], and the spatiotemporal dysregulation of cilia can therefore affect the development of many organ systems, including the central nervous, sensory, cardiovascular, digestive, metabolic, and skeletal systems [8,14,37,38]. These complex multisystem developmental disorders are collectively termed ciliopathies (Table 1).

### 2.1. Intellectual Disability

Defects in cerebral cortical development can lead to intellectual disabilities through a number of mechanisms [57]. Cortical development occurs in several steps: (1) Polarized radial glial progenitor formation, (2) radial/glial progenitor and intermediate progenitor proliferation, (3) radial/glial-guided neuronal migration, and (4) post-migratory neuronal differentiation, such as outgrowth and fasciculation of axons and dendrites [39]. The primary cilia of these progenitor cells play important roles in cerebral cortical development [40]. In Joubert syndrome, Bardet–Biedl syndrome (BBS), and oro-facial-digital syndrome, pathogenic mutations in genes regulating primary cilia function in progenitors disrupt cerebral cortical development [39], which can lead to intellectual disability [38].

### 2.2. Retinal Degeneration

The neuronal cell bodies of the retina are precisely organized in three major laminae: The ganglion cell, inner nuclear, and outer nuclear layers [58]. These layers are connected by neuronal projections of the cells located in each layer. Photoreceptor cells located in the outer nuclear layer capture photons and transmit signals to the brain through the inner nuclear and ganglion cell layers. Photoreceptor cells are composed of an outer segment (OS), inner segment (IS), and transition zone (also known as the connecting cilium), which connects the OS and IS [41,42]. The axoneme arises from the basal body in the IS and extends to the OS through the transition zone. Photon sensing is mainly performed by opsin proteins in the OS. Because the OS lacks protein synthesis machinery, opsin and other proteins involved in photon sensing are transported from the IS to the transition zone to the OS along the axoneme. Genes causative for ciliopathies, including Joubert syndrome, BBS, oro-facial-digital syndrome, Usher syndrome, and Meckel syndrome, are frequently involved in maintaining the structure and function of axonemes in photoreceptor cells [59]. Therefore, causative gene mutations in these ciliopathies often impair protein transport along the axoneme, resulting in retinal degeneration, and, potentially, vision loss, as is the case in retinitis pigmentosa and Leber congenital amaurosis [41,42,59].

### 2.3. Craniofacial Malformation

Fusion of distinct prominences, including the frontonasal, paired maxillary, and mandibular prominences, is crucial for proper craniofacial development [43]. Cranial neural crest cells (CNCCs) originate at the neural tube, undergo epithelial–mesenchymal transition, and migrate toward and proliferate in facial prominences [44]. The primary cilia of CNCCs play crucial roles in these steps through transduction of the Hedgehog and wingless-type MMTV integration site family (WNT) signaling pathways [43,44,60]. Gene mutations associated with ciliopathies affecting the function of primary cilia in CNCCs can lead to craniofacial malformations, such as cleft or lip palate, hyper/hypotelorism, micrognathia, and craniosynostosis [43,44,61,62].

### 2.4. Laterality Disorders

Although the human body is externally symmetrical, the visceral organs are arranged asymmetrically in a stereotyped manner [45,46,63]. Motile cilia of pit cells and nonmotile cilia of crown cells in the ventral node of the mammalian embryo play crucial roles in regulating left–right asymmetry [64,65]. When the motile cilia of pit cells generate leftward flow, the nonmotile cilia of crown cells located at the left side of the pit cells sense the flow and secrete Nodal–Gdf1 heterodimers [66,67]. In turn, the heterodimers bind to receptors in lateral plate mesoderm-derived cells and increase the expression of Nodal, Lefty2, and Pitx2, leading to left-side morphogenesis [68,69,70,71]. Accordingly, the impairment of cilia in the ventral node can cause laterality disorders, such as situs inversus and heterotaxy [45,46,72,73,74,75].

### 2.5. Cystic Kidney Disease

Cystic kidney disease is one of the main renal ciliopathies [36,76]. Renal tubular cells detect fluid flow through cilia. In these cells, fluid flow increases Ca^2+^ uptake through calcium channels, such as polycystin 2 [77,78], and the consequent increase in the intracellular Ca^2+^ concentration inhibits adenylate cyclase 6 and suppresses cyclic adenosine monophosphate (cAMP) signaling. The dysfunction of cilia in renal tubular cells prevents the increase in the Ca^2+^ concentration and suppression of cAMP signaling in response to fluid flow, resulting in activation of protein kinase A [47,48]. In turn, protein kinase A activation increases fluid secretion through chloride channels and deregulates multiple cellular pathways, including proliferation, apoptosis, and the polarity of renal tubular cells, leading to renal cystogenesis [79].

### 2.6. Obesity

Dysregulation of primary cilia in the central nervous system and peripheral tissues is associated with obesity, which often accompanies ciliopathies, such as BBS and Alström syndrome [49,50,80,81]. Obesity results from an excessive calorie intake relative to energy expenditure. In response to food intake, leptin is secreted from adipocytes and binds to receptors located in the primary cilia of anorexigenic and orexigenic neurons in the hypothalamus, resulting in increased and decreased expression of the anorexigenic peptide pro-opiomelanocortin and the orexigenic peptide Agouti-related peptide, respectively [82,83,84]. Genes associated with BBS and Alström syndrome regulate primary cilia in these hypothalamic neurons, and mutation of these genes can lead to obesity by failing to suppress the appetite through primary cilia-mediated leptin signaling [50].

Another contributing factor in obesity is elevated adipogenesis, resulting in an increased abundance of adipocytes [85]. Primary cilia are present in differentiating preadipocytes and play critical roles in adipogenesis [86]. Impairment of primary cilia in preadipocytes’ knockdown of BBS proteins (BBS10 and BBS12) stimulates adipogenesis by activation of the glycogen synthase kinase 3 pathway and nuclear accumulation of peroxisome proliferator-activated receptor γ [86]. Knockdown of BBS12 in human mesenchymal stem cells also impairs ciliogenesis and enhances adipogenesis [87]. In contrast, knockdown of intraflagellar transport 88 in preadipocytes or mesenchymal stem cells inhibits adipogenesis by impairing the localization of insulin-like growth factor-1 receptors in primary cilia [88,89]. The precise mechanisms by which primary cilia regulate adipogenesis remain to be fully elucidated.

### 2.7. Scoliosis

Scoliosis is a skeletal dysfunction characterized by abnormal spine curvature. Scoliosis is associated with an impaired structure and function of cilia [51]. The primary cilium of osteocyte acts as a hub in a mechanotransduction pathway for loading-induced bone adaptation [52,90]. In general, short primary cilia of osteocytes are perpendicularly oriented to the long axis of bone [91]. In contrast, primary cilia of osteoblasts from idiopathic scoliosis (IS) patients are significantly longer than those of control samples [51]. The induction of osteogenic factors, including bone morphogenic protein 2 (BMP2) and cyclooxygenase 2 (COX2), are impaired in osteoblasts from the IS patients and osteoblasts with elongated primary cilia by lithium chloride treatment [51]. These findings suggest that elongated primary cilia in osteocytes may be dysfunctional in mechanotransduction and warrant further investigation to elucidate the molecular mechanisms of scoliosis.

### 2.8. Respiratory Distress

Respiratory distress, which is characterized by congestion, coughing, tachypnea, and hypoxia, is a cardinal feature of primary ciliary dyskinesia (PCD) [53]. Motile cilia of the epithelial cell lining of most of the upper and lower respiratory tracts are dysfunctional in PCD [54,92,93]. The dysfunction of motile cilia causes the impairment of mucociliary clearance and mucus obstruction, resulting in bronchiectasis and respiratory failure [53]. Most PCD follow an autosomal recessive inheritance. The mutations identified as being causative of PCD explain roughly 70% of the affected individuals [93]. These mutations impair the structure and/or function of motile cilia by affecting the dynein arm, radial spoke, central apparatus, or multiciliation [93]. However, the clinical phenotype of PCD is highly variable [53]. The relationship between the genotype and clinical phenotype remains to be fully elucidated.

### 2.9. Infertility

The sperm tail has microtubule arrangement (9 × 2 + 2), which is similar to that of motile cilia [55]. Therefore, infertility is frequently observed in males with PCD [54]. There are some differences, however, between the sperm tail and motile cilia, including cell type-specific axonemal proteins and accessory structures specific to the sperm tail, such as the mitochondrial sheath, fibrous sheath, and outer dense fibers [55]. Mutations in genes causative of PCD are not always associated with male infertility and vice versa [55]. The impairment of the sperm tail affects sperm motility. The impairment of motile and primary cilia also affects the function of the reproductive tract in both males and females [54,56]. Structural and functional studies of cilia associated with infertility constitute an important area in reproductive research [55,56].

### 2.10. Roles of Primary Cilia in Cancer

Primary cilia in cultured mouse 3T3 fibroblasts and human retinal pigment epithelial (RPE1) cells can be disassembled and assembled by serum stimulation and deprivation, respectively [18,19,94]. Aurora A kinase (AURKA), one of the most important mitotic kinases for cell-cycle control [95], plays important roles in deciliation by serum stimulation [96,97]. AURKA is activated by serum stimulation through Ca^2+^/calmodulin signaling, the non-canonical WNT pathway, and phosphatidylinositol signaling [97,98,99,100]. Serum stimulation also activates AURKA through the pathway involving epidermal growth factor receptor (EGFR), ubiquitin-specific peptidase 8 (USP8), and trichoplein (TCHP) (described in the next section) [33,34,101]. Activated AURKA phosphorylates itself and target proteins during G1 phase, which stimulates the disassembly of primary cilia [97]. Several proteins associated with AURKA and ciliogenesis have been identified, including histone deacetylase 6 [94] and nudE neurodevelopment protein 1 (NDE1) [102]. In response to serum stimulation, NDE1 localizes at the basal body and suppresses ciliogenesis by tethering dynein light chain 1 [103]. Under serum deprivation conditions, cyclin-dependent kinase 5 is activated and phosphorylates NDE1. Phosphorylated NDE1 is then recognized and ubiquitylated by the E3 ligase complex SCF^FBXW7^, resulting in ciliogenesis [104,105]. Importantly, forced ciliation in cells growing under serum stimulation conditions can cause cell-cycle arrest [33,34,101,102,106]. These findings suggest that the primary cilium can act as a negative regulator of the cell cycle and may be a tumor suppressor organelle [3,4,5,6,9,107,108,109]. In fact, the suppression of primary cilia function is associated with tumorigenesis, cell proliferation, and metastasis in many cancers, including glioblastoma [110], esophageal cancer [111], colon cancer [112], cholangiocarcinoma [113,114], pancreatic ductal adenocarcinoma [115], clear cell renal carcinoma [116], prostate cancer [117], ovarian cancer [118,119], melanoma [120], and chondrosarcoma [121] (Table 2). However, primary cilia can promote tumor progression under certain conditions. In medulloblastoma and basal cell skin carcinoma caused by gain-of-function mutation of SMO, primary cilia convert the GLI transcription factors GLI2 and GLI3 to their activated forms, inducing their translocation to the nucleus, increased transcription of Hedgehog target genes, and promotion of cell proliferation [122,123]. In contrast, primary cilia of medulloblastoma and basal cell skin carcinoma caused by gain-of-function mutation of GLI2 increases the activity of GLI3 as a transcriptional repressor, resulting in suppression of the proliferation of these cancer cells [122,123]. Further work will thus be necessary to fully understand the context-dependent roles of primary cilia in cell proliferation.

## 3. Roles of E3 Ubiquitin Ligases and DUBs in Ciliogenesis, Ciliopathy, and Cancer

The structure and function of cilia are dynamically regulated by many ciliary proteins through posttranslational modification [2,4,5,6,25]. Ubiquitination and deubiquitination of ciliary proteins by E3 ubiquitin ligase and DUBs, respectively, are crucial for the dynamic regulation of cilia [4,5,6,31,32,33,34]. In this section, we briefly describe the role of E3 ubiquitin ligases and DUBs in ciliogenesis, ciliopathy, and cancer (Table 3).

### 3.1. CRL3^KCTD17^, USP8, and TCHP

TCHP, a centriolar protein originally identified as a keratin-binding protein, activates AURKA and suppresses ciliogenesis [101,141,142]. TCHP is ubiquitinated by the E3 ligase CRL3^KCTD17^, a complex of the scaffold protein Cullin 3, RING box protein 1 (RBX1), and potassium channel tetramerization domain-containing 17 (KCTD17) [33]. Knockdown of KCTD17 in RPE1 cells suppresses ciliogenesis by stabilizing TCHP, leading to the activation of AURKA [33]. NDE1-like 1 (NDEL1), a modulator of dynein activity localized at the subdistal appendage of the mother centriole [143,144], indirectly inhibits ubiquitination of TCHP by CRL3^KCTD17^ [106]. In contrast, TCHP is deubiquitinated by USP8 after EGFR-mediated phosphorylation of USP8 at tyrosine residues 717 and 810 [34]. Knockdown of USP8 in RPE1 cells induces ciliogenesis and cell-cycle arrest even in the presence of serum [34]. These findings suggest that forced ciliogenesis by inhibition of USP8 may be a potential therapeutic strategy for cancers with a high expression of USP8 and loss of cilia. In fact, USP8 is highly expressed and plays an oncogenic role in melanoma [124], and inhibition of USP8 suppresses the proliferation of glioblastoma stem cells [125]. The precise effect of USP8 on ciliogenesis in these tumor cells remains to be elucidated. In zebrafish, knockout of Usp8 increases ciliogenesis in renal tubules and causes renal cysts [34], whereas knockout of Kctd17 impairs ciliogenesis in Kupffer’s vesicle and causes situs inversus (Figure 1). Because AURKA is also associated with both cancer and ciliopathy [95,97], these findings suggest that the involvement of KCTD17 and USP8 in cancer and ciliopathy might be mediated by effects on ciliogenesis via a TCHP–AURKA pathway.

### 3.2. MARCHF7, TRIM32, USP9X, and IQCB1

The IQ motif containing B1 (IQCB1), also known as Nephrocystin-5, increases ciliogenesis by binding to centrosomal protein 290 (CEP290) [145]. IQCB1 is ubiquitinated by membrane-associated ring-CH-type finger 7 (MARCHF7) and the tripartite motif containing 32 (TRIM32, also known as BBS11) [126]. Overexpression of MARCHF7 or TRIM32 inhibits ciliogenesis [126]. MARCHF7 promotes proliferation and invasion of cervical cancer cells [127], and TRIM32 is also oncogenic in head and neck squamous cell carcinoma and skin cancer [128,129]. Conversely, IQCB1 is deubiquitinated and stabilized by USP9X [126]. Knockdown of USP9X inhibits ciliogenesis [126]. USP9X is a major tumor suppressor gene in pancreatic ductal adenocarcinoma [131]. These findings suggest that MARCHF7, TRIM32, and USP9X may be involved in cancer via the modulation of ciliogenesis. Mutation of TRIM32, USP9X, and their substrate IQCB1 causes various phenotypes related to ciliopathy [130,146,147].

### 3.3. CYLD and MIB1

Cylindromatosis (CYLD) is a member of the USP family of proteins and is expressed in centriolar satellites [148]. CYLD stimulates ciliogenesis by stabilizing centrosomal protein 70 (CEP70) and pericentriolar material 1 (PCM1) [132,133]. Stabilization of PCM1 by CYLD is mediated by deubiquitination of mindbomb E3 ubiquitin protein ligase 1 (MIB1), which is activated by ubiquitination on lysine 63 [133]. Lysine 63-ubiquitinated MIB1 then ubiquitinates PCM1 and stimulates its degradation in proteasomes. CYLD antagonizes the degradation of PCM1 by suppressing the activity of MIB1. MIB1 also ubiquitinates KIAA0586, a centrosomal protein also known as Talpid3, thereby stimulating its degradation and inhibiting ciliogenesis [136]. These findings suggest that CYLD and MIB1 positively and negatively, respectively, regulate ciliogenesis. Loss-of-function mutations in CYLD are associated with familial cylindromatosis, a condition involving multiple skin tumors [135]. MIB1 is oncogenic in upper urinary tract urothelial carcinomas [137]. Knockout or knockdown of CYLD in mice and zebrafish show phenotypes related to ciliopathy [134,149]. PCM1 and KIAA0586, both substrates of MIB1, are also associated with ciliopathies [149,150].

### 3.4. CRL2^VHL^

The tumor suppressor protein von Hippel–Lindau (VHL) is a component of an E3 ubiquitin ligase complex that also contains the scaffold protein Cullin 2 and RBX1 [151]. Mutations in VHL related to formation of the E3 ubiquitin ligase complex lead to von Hippel–Lindau syndrome, which can exhibit both ciliopathy and cancer phenotypes [140,152]. CRL2^VHL^ ubiquitinates the α subunit of the transcription factor hypoxia-inducible factor 1 (HIF1α), leading to its proteasomal degradation [138,153]. VHL and HIF1α positively and negatively, respectively, regulate ciliogenesis and cancer [107,139,140,154].

## 4. Future Directions

### 4.1. Identification of E3 Ubiquitin Ligases and DUBs Related to Cilia Assembly and Disassembly

As summarized above, the UPS has been implicated as a key system for the regulation of cilia assembly and disassembly [32,33,155]. Many E3 ubiquitin ligases and DUBs other than those listed in Table 3 have been identified as regulators of cilia assembly and disassembly. These include the E3 ubiquitin ligases cyclin F [156], FBW7 [104], NEDD4L [157], MYCBP2 [157], and UBR5 [158]; and the DUBs USP14 [159] and USP33 [160]; however, other enzymes undoubtedly remain to be identified. We showed that NDEL1, a modulator of dynein activity [143,144], inhibits ubiquitination of TCHP by CRL3^KCTD17^ and suppresses ciliogenesis in RPE1 cells incubated in the presence of serum [106]. Furthermore, in the absence of serum, NDEL1 is degraded by the UPS, resulting in the disappearance of TCHP from the mother centriole and induction of ciliogenesis [106]. The proteins involved in UPS-mediated NDEL1 degradation remain unknown.

One efficient approach to identifying E3 ubiquitin ligases of a substrate of interest is two-stepped global E3 screening, in which a wheat germ cell-free expression system is used to produce more than 1000 E3 ubiquitin ligases in the first step, and the enzymes are then screened using specific small interfering RNAs in the second step [33]. Genome-wide RNAi screening and proteomic profiling approaches may identify novel UPS proteins regulating cilia assembly and disassembly [157,161,162]. It will also be important to elucidate the subcellular compartment (basal body, transition zone, and/or axoneme) in which these E3 ubiquitin ligases and DUBs are active [32]. Compartment-specific proteomic profiling could be an efficient strategy to address these questions [163,164].

### 4.2. Identification of E3 Ubiquitin Ligase and DUB Substrates

Although many E3 ubiquitin ligases and DUBs involved in the assembly and disassembly of cilia have been identified, the precise substrates of many of them remain unknown. For example, we performed RNAi screening of RPE1 cells and identified six DUBs, USP8, USP38, USP43, USP52, USP54, and UCHL3, as suppressors of ciliogenesis [34]. We also identified TCHP as a substrate of USP8 in the regulation of ciliogenesis [34], but the substrates for the other five DUBs are unknown. A variety of experimental approaches have been developed to identify DUB substrates [165]. Stable overexpression or knockdown of DUBs followed by quantitative proteomic analysis to detect proteins differentially expressed in control and manipulated cells identified Sec28p and NFX1-123 as substrates of Ubp3p and USP9X, respectively [166,167]. Affinity purification proteomics using tagged DUBs has identified CEP192 as a substrate of CYLD [168]. Similarly, affinity purification proteomics using antibodies that recognize the diglycine residues, a remnant present on the ε-amine of lysine following trypsin digestion of ubiquitinated proteins, was successful in identifying S100A6 and hnRNP K as SseL substrates [169]. The diglycine remnant affinity purification method can also be used to identify E3 ubiquitin ligase substrates [170,171].

Once ligated to its substrate, ubiquitin itself can be modified by ubiquitination of one or more of its seven lysine residues (K6, K11, K27, K29, K33, K48, and K63) or the N-terminal methionine [172]. Polyubiquitination can be homotypic (same linkage) or heterotypic (different linkage) [173] and it plays various roles in cell signaling regulation depending on the linkage type [173]. For example, homotypic K48 polyubiquitination is related to classical proteasomal degradation; homotypic K63 polyubiquitination regulates protein–protein interactions; homotypic K6 and K27 polyubiquitinations are involved in the DNA damage response; and homotypic K29 and K33 polyubiquitinations are linked to innate immunity. In centriolar satellites, which are electron-dense and spherical cytoplasmic granules around centrosomes, modification of the E3 ubiquitin ligase MIB1 by homotypic K63 polyubiquitination induces homotypic K48 polyubiquitination of its substrate PCM1, resulting in PCM1 proteasomal degradation [133]. PCM1 plays an indispensable role in the clustering of centriolar satellites around the centrosome to orchestrate ciliogenesis [174]. CYLD located in centriolar satellites deubiquitinates the K63 polyubiquitin chain of MIB1, thereby antagonizing MIB1-mediated degradation of PCM1 and suppressing ciliogenesis [133]. Somatostatin receptor 3 (SSTR3) and G protein-coupled receptor 161 (GPR161) are important G protein-coupled receptors (GPCRs) that regulate somatostatin and hedgehog signaling, respectively, in primary cilia. K63 polyubiquitination of these GPCRs enables them to be recognized by ciliary exit machinery [175]. β-arrestin is known to mediate the K63 polyubiquitination of SSTR3 and GPR161, but the identity of the E3 ubiquitin ligase(s) involved is unclear. One important task for the future is the development of novel tools that will enable the identification of all E3 ubiquitin ligase and DUB substrates, as well as their ubiquitination patterns, related to the assembly and disassembly of cilia [172,173,176].

### 4.3. Identification of Drugs Targeting E3 Ubiquitin Ligases and DUBs

The screening of compounds for effects on protein ubiquitination and deubiquitination has led to the identification of a number of E3 ubiquitin ligase- and DUB-targeting drugs [29,165,177]. Because the substrates of these enzymes have a wide variety of functions, such drugs can interfere with various signaling pathways and impair physiological functions. Targeting the interaction of E3 ubiquitin ligases, DUBs, and their substrates involved in cilia assembly and disassembly may be a fruitful approach to developing selective drugs for the treatment of cancer and ciliopathies. Several technological advances have accelerated the development of drugs targeting protein–protein interactions [178,179]. The F-box protein S-phase kinase-associated protein 2 (SKP2), a component of E3 ubiquitin ligase SCF^SKP2^, ubiquitinates several proteins important for cell proliferation and survival, including p27^KIP1^, p21^CIP1^, and AKT serine/threonine kinases [180]. Some compounds have been identified that bind to a pocket in SKP2 that acts as the binding site for cyclin-dependent kinases regulatory subunit (CKS1), an accessory protein that can bind to p27^KIP1^ phosphorylated by cyclin-dependent kinase 2/cyclin E [181,182,183]. These chemicals inhibit the interaction between SKP2 and CKS1, resulting in selective inhibition of SKP2-mediated ubiquitination and degradation of p27^KIP1^. Screening for compounds that disrupt the subcellular translocation of E3 ubiquitin ligases and DUBs is another potential approach to developing selective drugs. USP4 and USP15 function in both the cytosol and the nucleus. In the cytosol, they deubiquitinate proteins involved in many signaling pathways, including those important to inflammation and oxidative stress; in the nucleus, they deubiquitinate proteins regulating splicing [184]. Nuclear translocation of these DUBs is inhibited by phosphorylation of two threonine residues located in the binding sites for spliceosome-associated factor 3 (SART3), a binding partner that facilitates the nuclear translocation of USP4 and USP15 [185]. Drugs interfering with the interaction between USP4 and USP15 and SART3 may selectively inhibit the DUB functions in the nucleus. Future work should include elucidation of the structure–activity relationships for E3 ubiquitin ligase- and/or DUB-targeting compounds and the identification of druggable sites in non-catalytic regions of the enzymes. Collectively, these approaches may lead to the development of novel drugs that regulate the enzymes in a context-dependent manner.

In summary, we have described the dysregulation of cilia in ciliopathies and cancers, and how that dysregulation results from changes in ciliary protein stability regulated by the UPS. As noted, some of the E3 ubiquitin ligases and DUBs involved in the maintenance of ciliary protein stability may be therapeutic targets for the associated disorders. Indeed, small molecules targeting these E3 ubiquitin ligases and DUBs, including USP8, USP9X, CYLD, and VHL, have been successfully developed [29,177,186]. However, the role of E3 ubiquitin ligases and DUBs in disease can be context dependent [28,148,187,188]. Thus, it will be important to develop small molecule modulators of the interactions between E3 ubiquitin ligases or DUBs and their binding proteins in a context-specific manner.

## Figures and Tables

**Figure 1 ijms-21-05962-f001:**
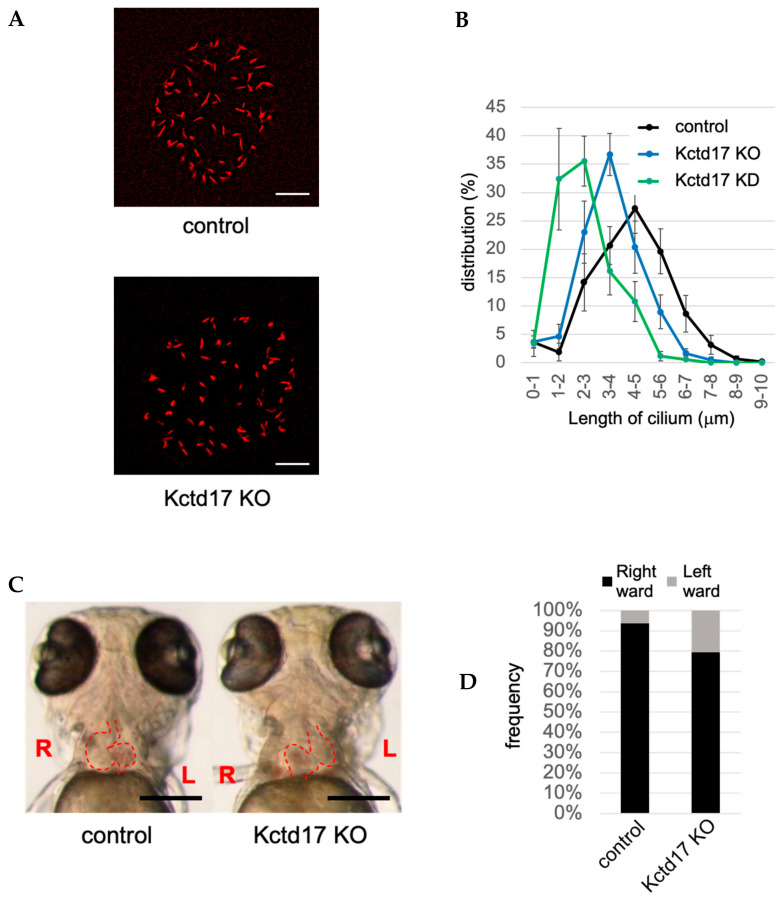
Suppression of Kctd17 in zebrafish impairs ciliogenesis in Kupffer’s vesicle and induces situs inversus (**A**) Staining of acetylated alpha-tubulin in the cilia of Kupffer’s vesicle at 12 h post-fertilization (hpf) of wild-type (control) or Kctd17 knockout (KO) zebrafish. (**B**) Distribution of the cilia length in Kupffer’s vesicle at 12 hpf in control or Kctd17 KO or knockdown (KD) zebrafish. (**C**) Representative in vivo images of control and Kctd17 KO zebrafish at 3 days post-fertilization (dpf). Control and Kctd17 KO zebrafish show rightward and leftward looping of the heart, respectively. Please see Appendix A. (**D**) At 3 dpf, rightward and leftward looping of the heart is observed in about 95% of the control zebrafish and about 20% of the Kctd17 KO zebrafish, respectively. Scale bar: 20 μm (**A**), 200 μm (**C**).

**Table 1 ijms-21-05962-t001:** The roles of cilia in ciliopathy phenotypes.

Ciliopathy Phenotype	Role of Cilia in the Phenotype	References
Intellectual disability	Dysfunction of cilia in radial glial progenitors impairs the proliferation, migration, and differentiation, resulting in the disruption of cerebral cortical development and intellectual disability.	[39,40]
Retinal degeneration	Mutation of genes related to the structure and function of axoneme in photoreceptor cells impair protein (e.g., rhodopsin) transport along the axoneme, resulting in retinal degeneration	[41,42]
Craniofacial malformation	Dysfunction of cilia in cranial neural crest cells impairs the epithelial-mesenchymal transition and the formation of facial prominences, causing craniofacial malformation such as cleft lip/palate	[43,44]
Laterality disorders	Dysfunction of cilia in ventral node fails to break left-right symmetry, left or right-side morphogenesis, causing laterality disorders, such as situs inversus and heterotaxy.	[45,46]
Cystic kidney disease	Dysfunction of cilia in renal tubular cells fails to detect fluid flow, increase Ca^2+^ concentration, and suppress protein kinase A, causing renal cystogenesis through dysregulated proliferation, apoptosis, and cell polarity.	[47,48]
Obesity	Dysfunction of cilia in hypothalamic neurons and adipocyte progenitor cess fails to suppress appetite and regulate appropriate differentiation to adipocytes, respectively, causing obesity.	[49,50]
Scoliosis	Primary cilia of osteoblasts are abnormally elongated and dysfunctional in mechanotransduction, which may impair loading-induced bone adaptation and cause scoliosis	[51,52]
Respiratory distress	Mutations of genes affecting dynein arm, radial spoke, central apparatus or multiciliation impair the structure and/or function of motile cilia of epithelial cells lining most of the respiratory tract, resulting in mucus obstruction and respiratory failure.	[53,54]
Infertility	Impairment of sperm tail, which has microtubule arrangement similar to that of motile cilia, cause sperm immotility and male infertility. Dysfunction of motile and primary cilia at the reproductive tract also causes both male and female infertility.	[55,56]

**Table 2 ijms-21-05962-t002:** The roles of primary cilia in cancer.

Cancer Cell	The Role of PC in the Cancer	References
Glioblastoma	Inhibition of HDAC6 restores the loss of PC and suppressed the proliferation	[110]
Esophageal squamous cell carcinoma	KD of PRDX1 restores the loss of PC and suppressed the proliferation	[111]
Colon cancer	Knockout of TTLL3 causes the loss of PC and promotes tumorigenesis in colon	[112]
Cholangiocarcinoma	The number of PC is frequently reduced. Inhibition of HDAC6 restores the loss of PC and suppressed the proliferation	[113,114]
Pancreatic ductal adenocarcinoma	Inhibition of HDAC2 in Panc1 induces ciliogenesis and suppressed the proliferation	[115]
Clear cell renal carcinoma	PC is lost by inactivation of VHL tumor suppressor	[116]
Prostate cancer	KD of TACC3 restores the loss of PC and suppressed the proliferation	[117]
Epithelial ovarian cancer	The number of PC is reduced, which is associated with centrosomal localization of AURKA. KD of AURKA restores the loss of PC and suppressed the oncogenic hedgehog signaling	[118,119]
Melanoma	Deconstruction of PC is sufficient to drive metastatic formation	[120]
Chondrosarcoma	Inhibition of HDAC6 restores the loss of PC and suppressed the proliferation	[121]
Medulloblastoma, basal cell carcinoma	-	-
with GOF mutation of SMO	PC increase transcriptional activator and stimulate proliferation	[122,123]
with GOF mutation of GLI2	PC increase transcriptional suppressor and inhibit proliferation	[122,123]

PC—primary cilia; KD—knockdown; GOF—gain-of-function.

**Table 3 ijms-21-05962-t003:** The roles of E3 ubiquitin ligases and deubiquitinases in ciliogenesis, ciliopathy, and cancer.

E3 Ligase or DUB	The Role in Ciliogenesis (Substrate)	The Role in Ciliopathy and/or Cancer	References
CRL3-KCTD17	inhibited by KD of KCTD17 (TCHP)	KO of KCTD17 causes situs inversus in zebrafish	[33], this paper (Figure 1)
USP8	stimulated by KD of USP8 (TCHP)	KO of USP8 causes cystic kidney in zebrafish	[34]
		USP8 is highly expressed and oncogenic in melanoma	[124]
		Inhibition of USP8 suppresses the proliferation of glioblastoma stem cells	[125]
MARCHF7	inhibited by OE of MARCHF7 (IQCB1)	MARCHF7 promotes proliferation and invasion of cervical cancer cells	[126,127]
TRIM32	inhibited by OE of TRIM32 (IQCB1)	TRIM32 is a causative gene of BBS (BBS11)	[126,127]
		TRIM32 is oncogenic in head and neck squamous cell carcinoma and skin cancer	[128,129]
USP9X	inhibited by KD of USP9X (IQCB1)	LOF mutations in USP9X cause phenotypes related to ciliopathy	[126,130]
		USP9X is a major tumor suppressor gene in pancreatic ductal adenocarcinoma	[131]
CYLD	inhibited by KD/KO of CYLD (CEP70, MIB1)	KO/KD of CYLD causes ciliopathy-related phenotype in mouse and zebrafish	[132,133,134]
		LOF mutation in CYLD cause skin cancer (familial cylindromatosis)	[135]
MIB1	inhibited by OE of MIB1 (PCM1, KIAA0586)	MIB1 is oncogenic in upper urinary-tract urothelial carcinomas	[133,136,137]
CRL2-VHL	stimulated by OE of VHL (HIF1A)	KO of VHL causes cystic kidney	[138,139]
		VHL is tumor-suppressive in renal cancers	[140]

DUBs—deubiquitinases; KD—knockdown; KO—knockout; OE—overexpression; LOF—loss-of-function.

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
