# Peer review of "Targeting E3 Ubiquitin Ligases and Deubiquitinases in Ciliopathy and Cancer"

_ijms, 2020, doi:10.3390/ijms21175962_

Round 1

Reviewer 1 Report

Dear Editor,

The review of Takashi Shiromizu, Mizuki Yuge, Kousuke Kasahara, Daishi Yamakawa, Takaaki Matsui, Yasumasa Bessho, Masaki Inagaki and Yuhei Nishimura « Targeting E3 ubiquitin ligases and deubiquitinases in ciliopathy and cancer» in a concise form provides a fairly complete information about the latest modern data on the structure, functioning and regulatory processes associated with cilia. I would like to note, however, that in the introductory part it is necessary to more accurately focus the reader's attention on the fact that the authors mainly describe the primary cilium. The structure and functioning of the mobile cilia and flagella has significant differences from the primary cilium. In general, the review corresponds to the subject matter of the journal and is written at a high scientific level, which, after a little revision, allows me to recommend it for publication in the International Journal of Molecular Science.

Minor points:

  • Line 28: It is necessary to give a general formula for the structure of motile cilia - (9x2 +2) (nine MT doublets and two central MT) and sensitive primary cilia - (9x2 +0) (nine MT doublets and 0 central MT). The primary cilium always grows on the mother centriole, in the case, when both centrioles form MT containing structures, as in spermatids, the motile flagellum grows on the daughter distal centriole, and the centriolar adjunct, morphologically more similar to the primary cilium and having the formula (9x2.5 + 0),  grows on the proximal mother centriole.  
  • Line 36: The cilium is  composed of three compartments: «The basal body-primary cilium complex is composed of three compartments:» Structure of motile cilia is more complex and divided to 5 different zones [1].  
  • Line 39: To add: MTs of cilia are prolongations of MT A and MT B of basal body.
  • Line 40: Ciliogenesis Primary cilia formation starts (Mechanism of ciliogenesis of motile cilia is different) It should be pointed out that the primary cilium exists in interphase cells and is disassembled before mitosis. [2]
  • Line 42-48: It is necessary to add references to publications in which the presented data was first shown.[3], [4], [5] for exemple.
  • Line 74: It should be emphasized that many ciliopathies are of a generalized nature - if the beating of the cilia in the respiratory epithelium is disturbed, then, with a high probability, the same defect can cause male sterility, due to disruption of the sperm flagellum movement activity. It is necessary to write about the existence of very short primary cilia in osteocytes [6]

Table 1

Information on the effects of ciliopathy on the functioning of the respiratory epithelium, oviduct epithelium and sperm motility should be added to the table and the text.

  1. Alieva, I.B.S., C.; Uzbekova, S.; Uzbekov, R.E. . A question of flagella origin for spermatids – mother or daughter centriole? . In Flagella and cilia. Types structure and functions, Uzbekov, R.E., Ed. Nova Science Publishers, Inc.: New York, 2018; pp 109–126.
  2. Rieder, C.L.; Jensen, C.G.; Jensen, L.C. The resorption of primary cilia during mitosis in a vertebrate (ptk1) cell line. Journal of ultrastructure research 1979, 68, 173-185.
  3. Craft, J.M.; Harris, J.A.; Hyman, S.; Kner, P.; Lechtreck, K.F. Tubulin transport by ift is upregulated during ciliary growth by a cilium-autonomous mechanism. The Journal of cell biology 2015, 208, 223-237.
  4. Malicki, J.; Avidor-Reiss, T. From the cytoplasm into the cilium: Bon voyage. Organogenesis 2014, 10, 138-157.
  5. Yadav, S.P.; Sharma, N.K.; Liu, C.; Dong, L.; Li, T.; Swaroop, A. Centrosomal protein cp110 controls maturation of the mother centriole during cilia biogenesis. Development 2016, 143, 1491-1501.
  6. Uzbekov, R.E.; Maurel, D.B.; Aveline, P.C.; Pallu, S.; Benhamou, C.L.; Rochefort, G.Y. Centrosome fine ultrastructure of the osteocyte mechanosensitive primary cilium. Microscopy and microanalysis : the official journal of Microscopy Society of America, Microbeam Analysis Society, Microscopical Society of Canada 2012, 18, 1430-1441.

Author Response

The review of Takashi Shiromizu, Mizuki Yuge, Kousuke Kasahara, Daishi Yamakawa, Takaaki Matsui, Yasumasa Bessho, Masaki Inagaki and Yuhei Nishimura « Targeting E3 ubiquitin ligases and deubiquitinases in ciliopathy and cancer» in a concise form provides a fairly complete information about the latest modern data on the structure, functioning and regulatory processes associated with cilia. I would like to note, however, that in the introductory part it is necessary to more accurately focus the reader's attention on the fact that the authors mainly describe the primary cilium. The structure and functioning of the mobile cilia and flagella has significant differences from the primary cilium. In general, the review corresponds to the subject matter of the journal and is written at a high scientific level, which, after a little revision, allows me to recommend it for publication in the International Journal of Molecular Science.

Line 28

It is necessary to give a general formula for the structure of motile cilia - (9x2 +2) (nine MT doublets and two central MT) and sensitive primary cilia - (9x2 +0) (nine MT doublets and 0 central MT). The primary cilium always grows on the mother centriole, in the case, when both centrioles form MT containing structures, as in spermatids, the motile flagellum grows on the daughter distal centriole, and the centriolar adjunct, morphologically more similar to the primary cilium and having the formula (9x2.5 + 0), grows on the proximal mother centriole. 

Line 36

The cilium is composed of three compartments:

«The basal body-primary cilium complex is composed of three compartments:»

Structure of motile cilia is more complex and divided to 5 different zones [1]. 

  1. Alieva, I.B.S., C.; Uzbekova, S.; Uzbekov, R.E. . A question of flagella origin for spermatids – mother or daughter centriole? . In Flagella and cilia. Types structure and functions, Uzbekov, R.E., Ed. Nova Science Publishers, Inc.: New York, 2018; pp 109–126.

Line 39

To add: MTs of cilia are prolongations of MT A and MT B of basal body.

We have revised the manuscript as suggested by the reviewer to clarify the structural difference between primary and motile cilia (lines 37-45).

Line 40

Ciliogenesis Primary cilia formation starts

(Mechanism of ciliogenesis of motile cilia is different)

It should be pointed out that the primary cilium exists in interphase cells and is disassembled before mitosis. [2]

  1. Rieder, C.L.; Jensen, C.G.; Jensen, L.C. The resorption of primary cilia during mitosis in a vertebrate (ptk1) cell line. Journal of ultrastructure research 1979, 68, 173-185.

Line 42-48

We have revised the manuscript to clarify the difference of ciliogenesis between primary and motile cilia (lines 37-45) and to include the reference suggested by the reviewer (lines 46-47).

It is necessary to add references to publications in which the presented data was first shown.

[3], [4], [5] for exemple.

  1. Craft, J.M.; Harris, J.A.; Hyman, S.; Kner, P.; Lechtreck, K.F. Tubulin transport by ift is upregulated during ciliary growth by a cilium-autonomous mechanism. The Journal of cell biology 2015, 208, 223-237.
  2. Malicki, J.; Avidor-Reiss, T. From the cytoplasm into the cilium: Bon voyage. Organogenesis 2014, 10, 138-157.
  3. Yadav, S.P.; Sharma, N.K.; Liu, C.; Dong, L.; Li, T.; Swaroop, A. Centrosomal protein cp110 controls maturation of the mother centriole during cilia biogenesis. Development 2016, 143, 1491-1501.

We have revised the manuscript to include the references suggested by the reviewer (lines 51, 53, and 55).

Line 74

It should be emphasized that many ciliopathies are of a generalized nature - if the beating of the cilia in the respiratory epithelium is disturbed, then, with a high probability, the same defect can cause male sterility, due to disruption of the sperm flagellum movement activity.

It is necessary to write about the existence of very short primary cilia in osteocytes [6]

  1. Uzbekov, R.E.; Maurel, D.B.; Aveline, P.C.; Pallu, S.; Benhamou, C.L.; Rochefort, G.Y. Centrosome fine ultrastructure of the osteocyte mechanosensitive primary cilium. Microscopy and microanalysis : the official journal of Microscopy Society of America, Microbeam Analysis Society, Microscopical Society of Canada 2012, 18, 1430-1441.

Table 1

Information on the effects of ciliopathy on the functioning of the respiratory epithelium, oviduct epithelium and sperm motility should be added to the table and the text.

We have revised the manuscript to show the roles of cilia in scoliosis, respiratory distress, and infertility and to include the reference suggested by the reviewer (Table 1 and lines 158-188).

Reviewer 2 Report

I found this is a nice and comprehensive review and I just have a few minor comments.

1) I think "Roles of primary cilia in cancer" (line 151) is better to be Section 2.7, “Roles of E3 ubiquitin ligases and DUBs in ciliogenesis, ciliopathy, and cancer” (line 184) should be the main title of Section 3, and the following subsections should be 3.1, 3.2 and so on. 

2) line 266-269: It is unclear how the two stepped screening works. I think the authors mean that the first screen tests >1000 E3 ligases for their ability to bind to substrates and then the second screening tests functions of the identified E3 ligases by RNAi.

3) line 22: I think it is not necessary to use "may be" and "potential" in a same sentence. It can be either "are potential therapeutic targets" or "may be therapeutic targets"

Author Response

I found this is a nice and comprehensive review and I just have a few minor comments.

1) I think "Roles of primary cilia in cancer" (line 151) is better to be Section 2.7, “Roles of E3 ubiquitin ligases and DUBs in ciliogenesis, ciliopathy, and cancer” (line 184) should be the main title of Section 3, and the following subsections should be 3.1, 3.2 and so on.

We have revised the manuscript as suggested by the reviewer (lines 189, 221, 228, 256, 267, and 282)

2) line 266-269: It is unclear how the two stepped screening works. I think the authors mean that the first screen tests >1000 E3 ligases for their ability to bind to substrates and then the second screening tests functions of the identified E3 ligases by RNAi.

We have revised the manuscript as suggested by the reviewer to clarify the first and second steps (lines 304-305).

3) line 22: I think it is not necessary to use "may be" and "potential" in a same sentence. It can be either "are potential therapeutic targets" or "may be therapeutic targets"

We have revised the manuscript as suggested by the reviewer (line 22).